# How do social network models compare to all-to-all models for forecasting tuberculosis epidemics? A mathematical modeling study

Masabho P. Milali[1]*, Hae-Young Kim[1], George F. Corliss[2], Anna Bershteyn[1]

1 Department of Population Health, NYU Grossman School of Medicine, New York, New York, United States of America, 2 Department of Electrical and Computer Engineering, Marquette University, Milwaukee, Wisconsin, United States of America

* masabho.milali@nyulangone.org

## Abstract

### Background

Mathematical models guide tuberculosis (TB) target-setting, yet most assume homogeneous "all-to-all" mixing. We compared projected intervention impacts between an all-to-all compartmental model and a Barabási–Albert (BA) scale-free social network model under otherwise identical disease assumptions.

### Methods

We calibrated transmission parameters so both models produced similar baseline trends, then introduced vaccination (coverage 30–70%; efficacy 80–95%) and treatment (20–50% increases in recovery) after a 400-day burn-in. Outcomes were assessed 300 days post-intervention.

### Results

Under 60% coverage, increasing vaccine efficacy from 80% to 95% yielded smaller projected reductions in active TB with the network model than with all-to-all mixing. Treatment improvements showed the same pattern: lower reductions under the network than the all-to-all model at modest efficacy, converging at high efficacy/coverage. Findings were robust across baseline prevalence scenarios.

### Conclusions

Accounting for social networks can attenuate projected impacts for sub-optimal TB interventions. Forecasts and target-setting should include sensitivity to social network structure.

**Data availability statement:** All relevant data are within the paper.

**Funding:** This study was supported by the Section for Global Health, Department of Population Health, NYU Grossman School of Medicine in the form of institutional support awarded to AB, and by the Division of Comparative Effectiveness & Decision Science, Department of Population Health, NYU Grossman School of Medicine in the form of salary support for MPM, HK, and AB. The specific roles of these authors are articulated in the 'author contributions' section. The funders had no role in study design, data collection and analysis, decision to publish, or preparation of the manuscript.

**Competing interests:** he authors have declared that no competing interests exist.

## Introduction

Tuberculosis (TB) is a communicable disease caused by *Mycobacterium tuberculosis* complex bacteria and remains one of the leading infectious causes of death worldwide [1,2]. TB manifests in different forms, broadly classified by organ involvement (pulmonary versus extrapulmonary), drug sensitivity (drug-sensitive TB, multidrug-resistant TB [MDR-TB], and extensively drug-resistant TB [XDR-TB]), and disease activity (latent versus active TB) [3,4]. Pulmonary TB is the most common and the main driver of transmission, while extrapulmonary TB, more frequent among immuno-compromised individuals and children, can also contribute to morbidity and mortality [5]. Drug-sensitive TB is generally curable with standard first-line regimens (treatment success ~85%), whereas MDR-TB and XDR-TB require longer, more toxic, and cost-lier therapies, and are associated with substantially lower treatment success rates (~56% and ~39%, respectively) [6].

Globally, TB continues to impose a heavy burden, with both drug-sensitive and drug-resistant forms threatening progress toward elimination goals. In response to this high burden, the World Health Organization (WHO) has established the End TB strategy, which aims to reduce TB mortality by 95% and new cases by 90% from 2015 to 2035 [7,8]. Unfortunately, despite efforts to scale up interventions, TB incidence has not decreased to the expected levels, and global trends remain far off track from achieving WHO targets, with persistent transmission in both high-burden settings, where prevalence and incidence are elevated, and low-burden settings, where transmission is more sporadic but can still occur in high-risk subpopulations [9,10]. Vaccination (currently limited to BCG, with modest and age-dependent efficacy) and treatment have been central to TB control, with notable success in reducing mortality, but their population-level impact is constrained by coverage, adherence, and efficacy—particularly in the context of MDR-TB [9]. Beyond BCG, new vaccine candidates are in clinical development, but their eventual availability, timing of rollout, and uptake remain uncertain, and these factors are expected to influence their population-level impact critically [11]. Previous modeling studies have shown that delayed or staggered vaccine introduction can markedly reduce projected effectiveness compared with instantaneous implementation [11].

Patterns of TB transmission differ between high- and low-endemicity settings. In high-burden regions, sustained community transmission is driven by factors such as overcrowding, HIV co-infection, and limited healthcare access. In contrast, in low-burden settings, outbreaks are often linked to specific risk groups such as incarcerated populations, migrants, or people experiencing homelessness. Across all contexts, mathematical models have been essential for characterizing transmission dynamics, identifying key drivers, and informing intervention strategies [12]. These models suggest that TB incidence could fall substantially with increases in diagnosis, treatment, and prevention [13–15].

Conventional TB models assume all-to-all mixing. An infected individual has an equal chance of exposing everyone in the population or a pre-specified subgroup of the population (e.g., a particular age group). An important assumption implicit in these models is that social networks, which dictate that a given individual has contact

with only specific other individuals, are negligible in estimating the impact of TB interventions. An alternative modeling approach is to model transmission on social networks explicitly. Barabási-Albert scale-free social networks are often used for disease transmission modeling because they closely match measured social contact patterns from infectious disease studies [16–20]. These networks follow a power-law distribution of the number of social contacts per person, allowing for heterogeneity in the number of contacts, so that individuals with large numbers of contacts act as "super-spreaders"– i.e., those who disproportionately contribute to transmission [19–22].

Though less common than "all-to-all" modeling approaches, some studies have used social network models to study the dynamics of TB transmission and assess intervention strategies [23–26]. However, to our knowledge, no study has directly compared "all-to-all" versus social network-based models, holding constant as many aspects of the model structure and assumptions as possible while producing the same baseline epidemic trends from each model. One study of whooping cough compared "all-to-all" and social network models, concluding that a social network model provided more accurate forecasts of cases compared to an "all-to-all" model [27]. Because the global burden of TB is far higher than that of whooping cough, it is important to investigate whether modeling social networks influences the projected impact of TB interventions.

In this study, we systematically compared the impact of TB interventions in a social network model and an all-to-all model. The natural history of the disease and most intervention parameters were informed by published literature [28,29], with only the proportion of susceptible individuals receiving vaccination and the rate of recovery following treatment assumed. We applied identical assumptions regarding TB progression and intervention efficacy across both models to isolate the influence of contact structure on projected outcomes. This simplification was intentional, recognizing that real-world epidemics may not evolve identically under homogeneous versus network-based mixing.

We hypothesized that both an "all-to-all" and a social network model could produce similar TB epidemic patterns. We further hypothesized that the social network model, which confers more intense exposure to a smaller number of social contacts, would forecast lower decrease in TB prevalence compared to an "all-to-all" model because intensely-exposed contacts would become infected despite the benefits of treatment or prevention.

## Materials and methods

This study is reported in accordance with EPIFORGE 2020 guidelines [30].

### Study overview and objectives

This modelling study compared projections from two distinct TB transmission frameworks: an all-to-all compartmental model and a Barabási–Albert (BA) scale-free social network model. In a scale-free social network model, a small number of nodes act as hubs with disproportionately many connections, while most nodes have few connections (Fig 1).

This heterogeneity reflects realistic social contact patterns more accurately than homogeneous mixing assumptions. The models represented all-form tuberculosis, without stratification by organ involvement or drug resistance, because the analysis was conducted at an overall population level rather than a subgroup-specific level. The objective was to isolate the influence of contact network structure on projected intervention impacts, while keeping TB natural history, epidemiological parameters, and intervention assumptions constant across models.

### Model structure and assumptions

In both models, the population was stratified into susceptible (S), exposed (E), latent (L), infectious (I), and vaccinated (V) compartments, adopting a Susceptible-Exposed-Latent-Infectious (SELI) structure with an added vaccination compartment.

Susceptible individuals could become exposed following contact with infectious individuals at rate $\beta IS$, entering the exposed compartment (E), or receive vaccination through a one-time mass campaign at the time of intervention, in which

# A

# B

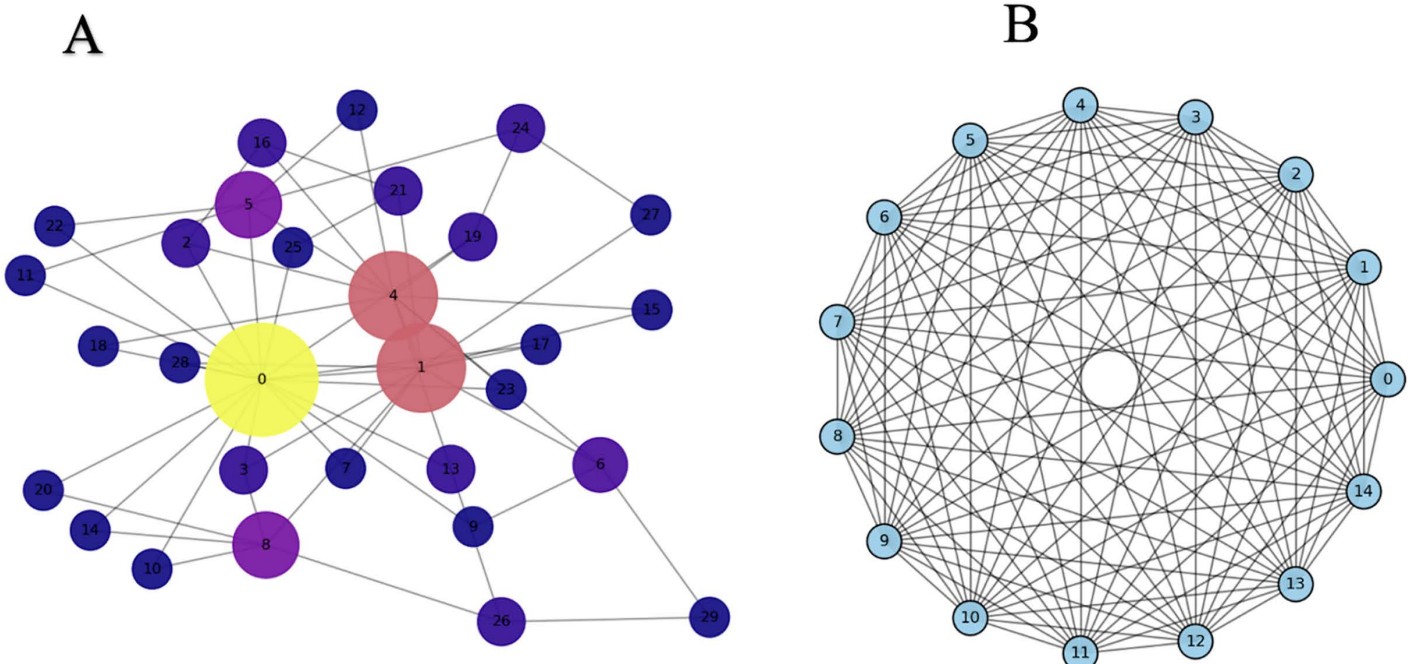

**Fig 1. Illustrative network models generated in Python using the NetworkX package.** (A) Barabási–Albert (scale-free) social network and (B) all-to-all social network. Each number represents an individual node (e.g., a person or agent in a social network). The size of each circle indicates how many connections that node has. In the scale-free network (A), some nodes have many more connections than others, for example, Node 0 has a very high number of links (large circle), while Node 17 has only a few (small circle). In contrast, in the all-to-all network (B), every node is connected to every other node, so all nodes have the same number of connections.

a fraction λ of S was transferred to V. Exposed individuals progressed either to latent infection at rate $(1–p)σE$ or directly to active disease (I) at rate $pσE$. Latently infected individuals could reactivate to become infectious at rate $γL$. Thus, TB cases (infectious cases) refer to individuals entering the infectious (I) compartment, either from the latent (L) compartment (reactivation) or directly from the exposed (E) compartment (primary progression).

Infectious individuals could recover at rate $δ(t)I$ and return to S. The recovery term $δ(t)I$ represents successful treatment of active TB, with recovered individuals re-entering the susceptible pool since no separate recovered (R) compartment was modeled.

Infectious individuals experienced both TB-related mortality (ω) and natural mortality (μ), while all other compartments experienced only natural mortality (μ). All deaths exited the system into a death sink (D), while births replenished S, ensuring constant population size. This constant-population assumption was a deliberate simplification to ensure comparability between frameworks; in reality, demographic changes could occur, but their impact is expected to be modest relative to TB transmission dynamics.

Vaccination was modeled as providing complete protection against infection, reflecting a hypothetical future vaccine with sterilizing immunity. In this scenario, vaccinated individuals could not become infected or transmit TB. In real-world settings, vaccines are imperfect, and breakthrough infections can occur; however, these were not represented here. This assumption was chosen to highlight the influence of network structure on intervention impacts without additional complexity from waning immunity or breakthrough infections.

In the BA model, a small proportion of nodes ("hubs") had disproportionately high connectivity, capturing heterogeneity in contact patterns observed in real-world populations. In contrast, the all-to-all model assumed homogeneous mixing, with every susceptible individual equally likely to contact any infectious individual.

The compartmental structure and transitions for both models are illustrated in **Fig 2**.

The modelled population was not representative of any specific setting or geographic location and was not stratified into age, risk, or socio-demographic subgroups such as children under five years, older adults, immunocompromised individuals (including those with HIV/AIDS), incarcerated populations, migrants, or other high-risk groups. The simulated population represented a generic adult, mixed-age cohort without an explicit age structure.

The baseline scenarios assumed a population size of 100,000 individuals, initialized with 99,990 susceptible individuals and 10 active TB cases. The BA network model had an average network density of 12 contacts per person, with a range from 6 to 1,035 contacts per person. In sensitivity analyses, we varied the network density to 20, 40, and 60 contacts per person and repeated the simulations ten times with different random seeds to account for the stochastic nature of the social network model (Fig 3).

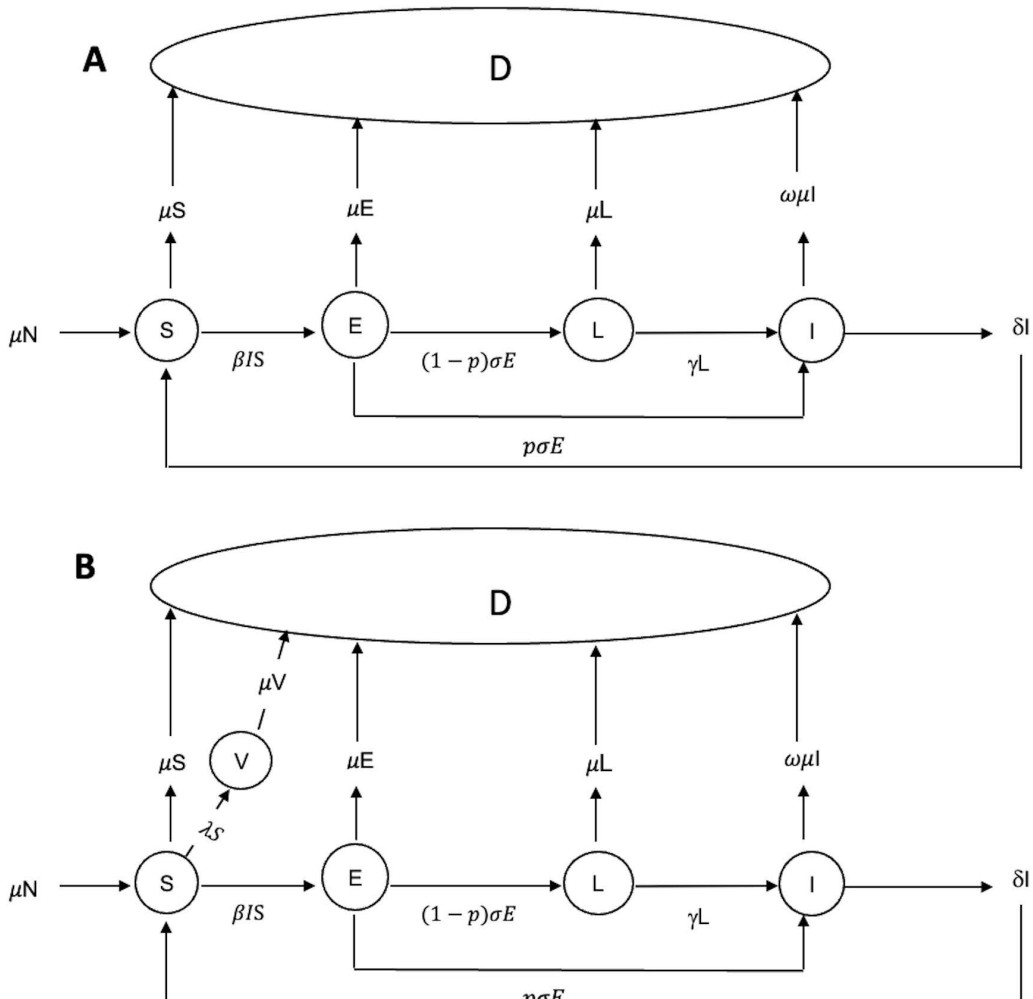

**Fig 2. Model diagram (A) before interventions are applied, and (B) after interventions are applied.** In addition to these flows, a fraction λ of the susceptible population S is transferred to the vaccinated state V at the time of the intervention.

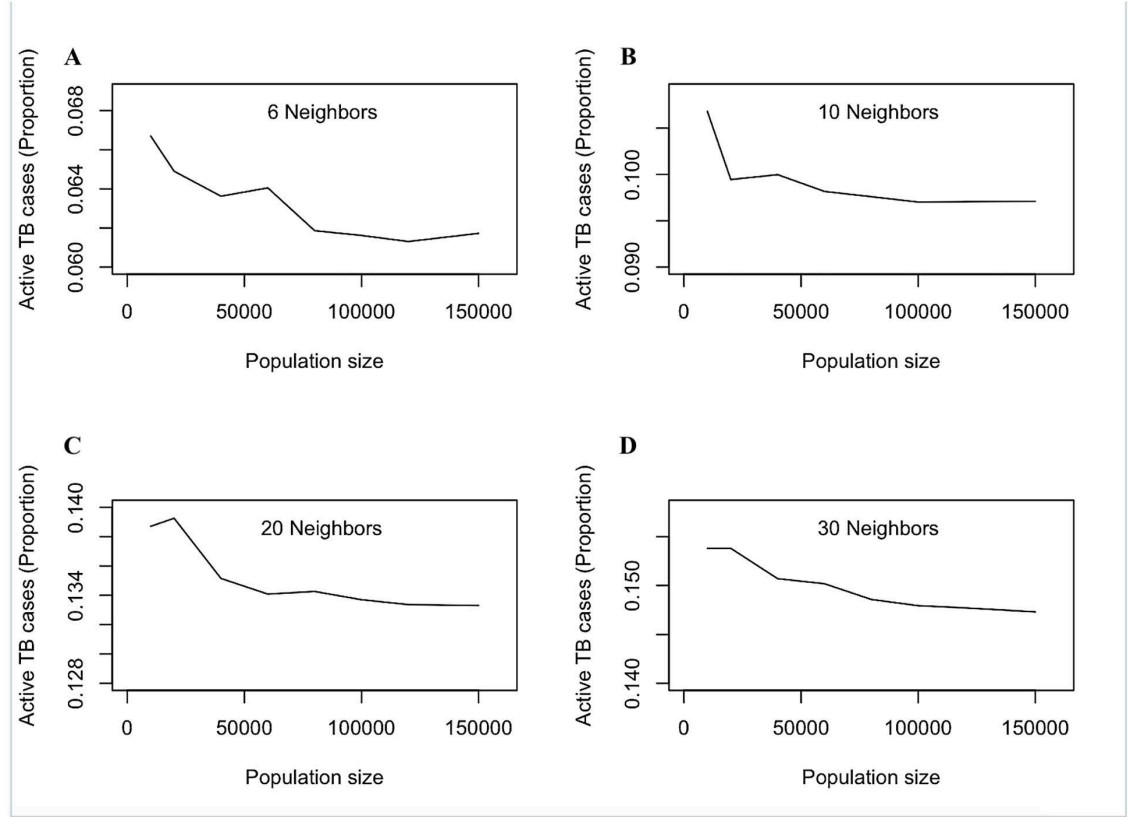

**Fig 3. Sensitivity of active TB prevalence to population size under varying average numbers of social contacts in the Barabási–Albert network model.** (A) 6 neighbors, (B) 10 neighbors, (C) 20 neighbors, and (D) 30 neighbors. Each panel shows the equilibrium proportion of active TB cases as the simulated population size increases. Higher average connectivity leads to greater baseline transmission intensity, resulting in higher equilibrium prevalence. Across all contact-degree settings, prevalence stabilizes as population size becomes sufficiently large, demonstrating that the model yields consistent large-population behavior despite network stochasticity.

## Model equations

The model dynamics were represented by the following system of ordinary differential equations (ODEs), formulated following standard approaches in TB modelling [28]:

$$\frac{dS}{dt} = \mu N - \beta IS - \lambda(t)S - \mu S + \delta(t)I \tag{1}$$

$$\frac{dV}{dt} = \lambda(t)S - \mu V \tag{2}$$

$$\frac{dE}{dt} = \beta IS - p\sigma E - (1-p)\sigma E - \mu E \tag{3}$$

$$\frac{dL}{dt} = (1-p)\sigma E - \gamma L - \mu L \tag{4}$$

$$\frac{dI}{dt} = p\sigma E + \gamma L - \mu I - \omega I - \delta(t)I \tag{5}$$

with intervention functions are defined as:

$$\delta(t) = \begin{cases} \delta_0, & t < t_{intervention}, \\ \delta_x, & t \geq t_{intervention} \end{cases} \tag{6}$$

$$\lambda(t) = \begin{cases} 0, & t < t_{intervention}, \\ \infty, & t = t_{intervention}, \text{ s.t. a fraction } \lambda \text{ of state S is transferred to state V at this time.} \\ \lambda, & t > t_{intervention} \end{cases} \tag{7}$$

## Definitions of terms

In this study, incidence refers to the number of new active TB cases per unit time; prevalence refers to the proportion of the population with active TB at a given time. Treatment recovery rate is the proportion of treated active TB cases that transition to the recovered state per unit time; treatment efficacy is the proportion of treated individuals who achieve bacteriological cure under full adherence; treatment adherence is the proportion of individuals completing the prescribed regimen; imperfect adherence refers to any deviation from completing the prescribed treatment regimen. Vaccine efficacy is the proportional reduction in susceptibility among vaccinated individuals, and vaccine coverage is the proportion of the susceptible population vaccinated. Steady-state (equilibrium) refers to a stable epidemiological condition in which the incidence and prevalence of TB remain constant over time, in the absence of new interventions, following an initial transient period.

## Parameterization and calibration

We used the same parameters for both models, except for the transmission rate from an infectious host to a susceptible host (β). The all-to-all model used a per-capita transmission rate (β_all) per day derived from published estimates, while the BA network model used a per-contact transmission rate (β_net) per day (Table 1). To ensure that both models produced comparable epidemic trajectories for incidence and prevalence in the absence of interventions, β_net was

**Table 1. Parameters used in both social network and all-to-all models. The value of β_net was obtained by adjusting β in the social network model until the prevalence of TB disease matched that of the all-to-all model. The resulting β_net was higher because it represents the force of infection for direct contacts of a case, rather than for the total population.**

| Symbol | Definition | Parameter | Units | Source |
|--------|-----------|-----------|-------|--------|
| $\beta\_all$ | Transmission rate in the all-to-all transmission model | 0.00000767 | Per person per day | [28] |
| $\beta\_net$ | Transmission rate in the network transmission model | 0.048 | Per person per day | Calibrated |
| $\lambda$ | Proportion of the susceptible population getting vaccinated | 0.3, 0.4, 0.6, 0.7 | Unitless | Assumed |
| $\mu$ | Birth rate/Death rate | 0.0008 | Per day | [29] |
| $\delta_0$ | Rate of recovering from TB before treatment | 0.02 | Per day | [28] |
| $\delta_x$ | Rate of recovering from TB after treatment | 0.2, 0.3, 0.4, 0.5 | Per day | Assumed |
| $\sigma$ | Rate of leaving exposed state | 0.2 | Per day | [28] |
| $\gamma$ | Rate of latent persons becoming active TB cases | 0.2 | Per day | [28] |
| $p$ | Proportion of exposed persons becoming active TB cases | 0.075 | Unitless | [28] |
| $\omega$ | Rate of dying from TB | 0.5 | Per day | [28] |

calibrated to $4.8 \times 10^{-2}$ per day, while β_all was fixed at the published per-capita value of $7.67 \times 10^{-6}$ per day (Fig 4). Because the two models represent contact structures differently, β_all and β_net are not directly comparable as parameter values; calibration was defined by aligning epidemic trajectories (incidence and prevalence) between the models rather than by equating parameter magnitudes.

Calibration was performed to align baseline prevalence and incidence with typical values reported for high-burden settings. External validation was not performed due to the generic, non-setting-specific nature of the models; limitations of this are noted in the Discussion.

## Intervention scenarios

The assumption of a constant population size (births balancing deaths) was maintained throughout the 700-day simulation horizon to ensure comparability between models. After a 400-day burn-in period, allowing the models to reach equilibrium (the point at which state variables stabilize over time in the absence of new inputs or interventions), two types of interventions were simulated: vaccination and treatment (Fig 4).

Vaccination was assessed at coverage levels of 30%, 40%, 60%, and 70%, with vaccine efficacy set at 100%. It was modelled using a generic vaccine profile to enable scenario exploration and did not explicitly represent BCG or other candidate vaccines. We acknowledge that BCG remains the only licensed TB vaccine, with limited efficacy in adults and

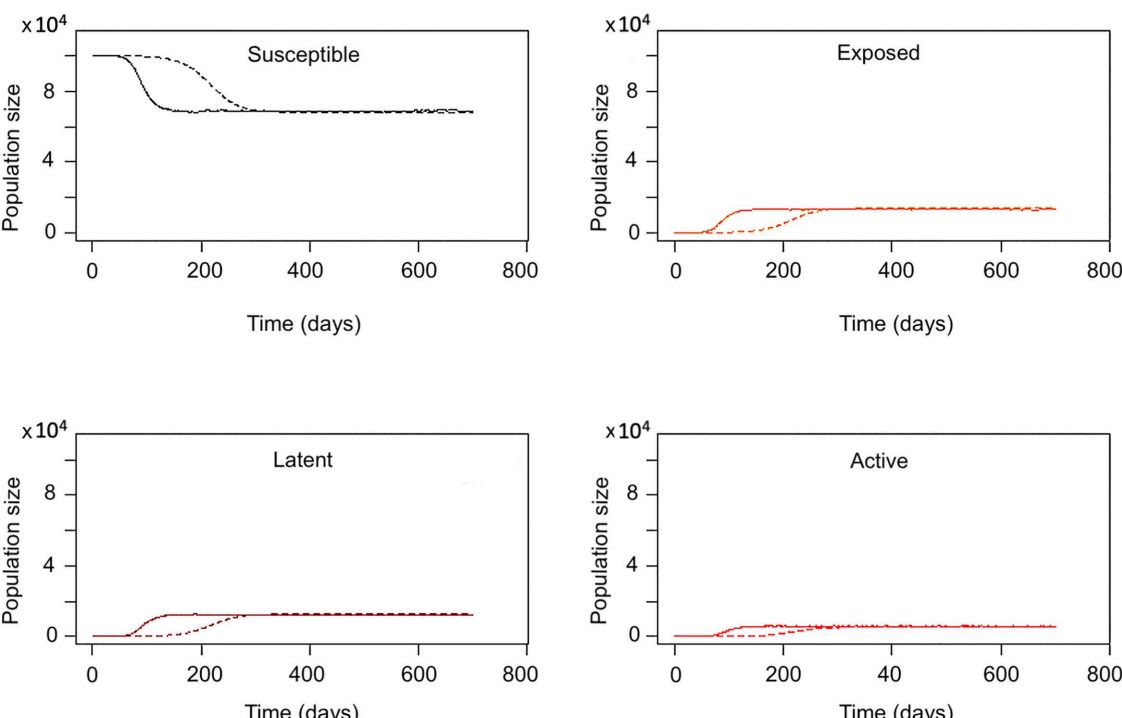

**Fig 4. Baseline compartmental dynamics of the all-to-all (dashed line) and social network (solid line) models prior to intervention.** Time-series trajectories show the population sizes in each disease state: (A) Susceptible, (B) Exposed, (C) Latent, and (D) Active TB. Both models begin with identical initial conditions and converge toward equilibrium, but differences in contact structure yield distinct transitional dynamics. The network model exhibits faster early depletion of susceptible individuals and earlier increases in exposed and latent cases, reflecting concentrated transmission within highly connected nodes, whereas the all-to-all model shows more gradual transitions. By approximately day 280, both models stabilize at comparable equilibrium levels across all compartments.

minimal use in low-endemicity settings; this was not incorporated into the current modelling approach, which was not specific to any geographic setting.

Treatment was modelled as a generic intervention that increased the recovery rate from active TB by 20%, 30%, 40% or 50% over baseline, representing exploratory improvements in programmatic effectiveness rather than strain-specific (drug-sensitive or drug-resistant) changes. It was represented as a composite regimen with efficacy assumptions aligned with WHO-reported programmatic outcomes for tuberculosis.

Coverage and efficacy levels for both treatment and vaccination were selected to span plausible ranges reported in previous TB modelling and empirical studies [29,30]. Intervention impacts were then evaluated 300 days after introduction (day 700 of simulation) to capture medium-term effects while avoiding excessive overlap with longer-term stochastic drift. The impacts were assessed by comparing infectious prevalence trajectories under intervention with those from a no-intervention baseline.

### Outcome measures

The primary outcome was the percent reduction in the number of infectious individuals at 300 days post-intervention (day 700) compared with a no-intervention baseline. Secondary outcomes included the time series trajectory of infectious prevalence under each scenario and across vaccine efficacy and baseline prevalence sensitivity analyses.

### Analysis methods

Both all-to-all and social network models were implemented deterministically using R version 4.0.4. The all-to-all model employed the 'deSolve' 1.28 package [26] for numerical integration of the system of ordinary differential equations. The Barabási–Albert social network model was implemented using the 'fastnet' 1.0.0 package [27] for network generation and analysis, together with the 'matrix' 1.3–2 package [28] for efficient handling of large adjacency matrices.

### Ethical considerations

This study used only simulated data and did not involve human participants, animals, or identifiable personal information. We complied with applicable ethical standards for modeling research.

### Reproducibility and data availability

All model equations and parameter values are provided within this manuscript. The R code used to generate all results is publicly available at https://github.com/masabho/TB_ScaleFree_SocialNetwork for full reproducibility and independent validation of the findings.

## Results

Interventions in both models were introduced after a 400-day burn-in period, and their impacts were evaluated over the subsequent 300 days (day 700 of the total simulation). Overall, both models agreed that higher vaccine coverage and efficacy, as well as improved treatment recovery rates, substantially reduced the prevalence of infectious individuals. However, the BA model consistently estimated smaller gains at lower intervention intensities due to heterogeneity in social connectivity.

### All-to-All model results

In the all-to-all model, vaccination produced strong, graded reductions in active TB that scaled with both coverage and efficacy. At 30% coverage, assuming 100% vaccine efficacy, the reduction in infectious prevalence after 300 days was 86±1.3%. At 60% coverage, reduction increased to 96±0.9%, and by 70% coverage the reduction approached 99±0.3% (Fig 5A). Similarly, treatment scale-up produced proportional reductions in active TB. A 20% increase in recovery rate resulted in an 82±0.9% reduction, while a 50% increase produced near-complete elimination of cases (100±0.1%) (Fig 5B).

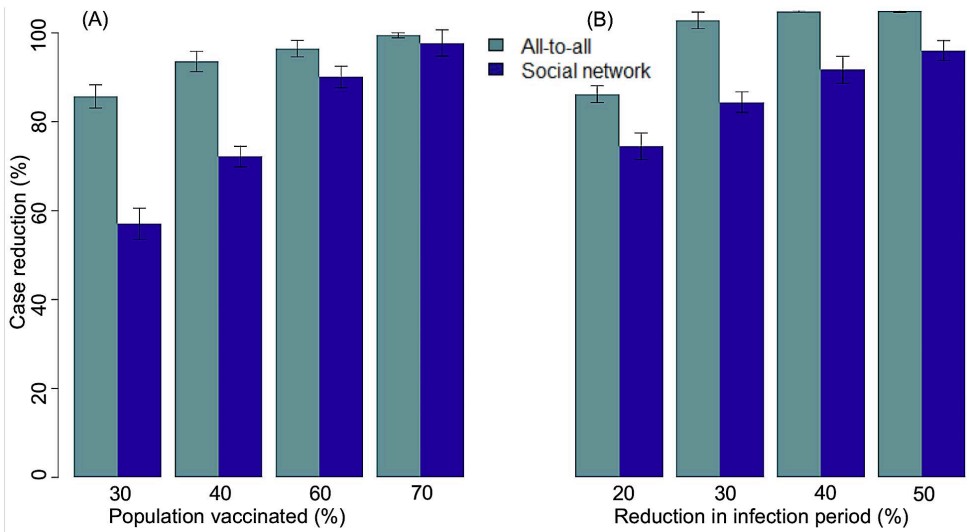

**Fig 5. Comparison of intervention impact between all-to-all model (light blue) and social network model with an average of 12 contacts per person (dark blue).** (A) Percent reduction in active TB cases following vaccination with 100% efficacy administered to 30%, 40%, 60%, and 70% of the population. (B) Percent reduction in active TB cases when treatment improves the recovery rate by 20%, 30%, 40%, and 50% over baseline. Error bars indicate variability across repeated simulations in the models. Across both intervention types, the social network model consistently projects smaller reductions at lower coverage or weaker treatment effects, reflecting concentrated transmission within highly connected individuals.

Time-series dynamics showed steep declines in infectious prevalence immediately after intervention onset, followed by stabilization at lower plateaus by day 700. Higher coverage levels (60–70%) produced rapid reductions that nearly eliminated transmission, while lower coverage (30–40%) produced slower decreases but still achieved substantial reductions by day 700 (Fig 6).

### Barabási–Albert (BA) scale-free network model results

In the BA social network model (baseline average network degree = 12 contacts per person), intervention effects were generally less pronounced than in the all-to-all framework, particularly at lower coverage or treatment effect sizes. At 30% vaccination coverage, case reduction was markedly lower (57 ± 1.8% vs. 86 ± 1.3% in the all-to-all model). At 60% coverage, the difference narrowed (90 ± 1.2% vs. 96 ± 0.9%), and by ≥70% coverage the models converged (96–98%) (Fig 5A). Treatment improvements also yielded smaller relative benefits in the BA model: a 20% increase in recovery produced a 71 ± 1.4% reduction compared with 82 ± 0.9% in the all-to-all model, while a 50% increase achieved 91 ± 1.1%, still below the near-elimination scenario of the all-to-all framework (Fig 5B).

Time-series dynamics showed sharper early declines in infectious prevalence after intervention onset compared with the all-to-all model, but the curves flattened sooner, leaving a persistent plateau by day 700. At lower vaccination coverage (30–40%), divergence from the all-to-all model was most evident, with the network model sustaining higher levels of infection. At higher coverage (60–70%), the trajectories converged more closely, although small residual infection remained due to highly connected hubs in the BA network (Fig 6).

### Sensitivity analysis results

Sensitivity analysis confirmed that increasing the average number of contacts per person in the BA network (from the baseline of 12–20, 40, or 60) led to higher baseline prevalence of TB, as expected.

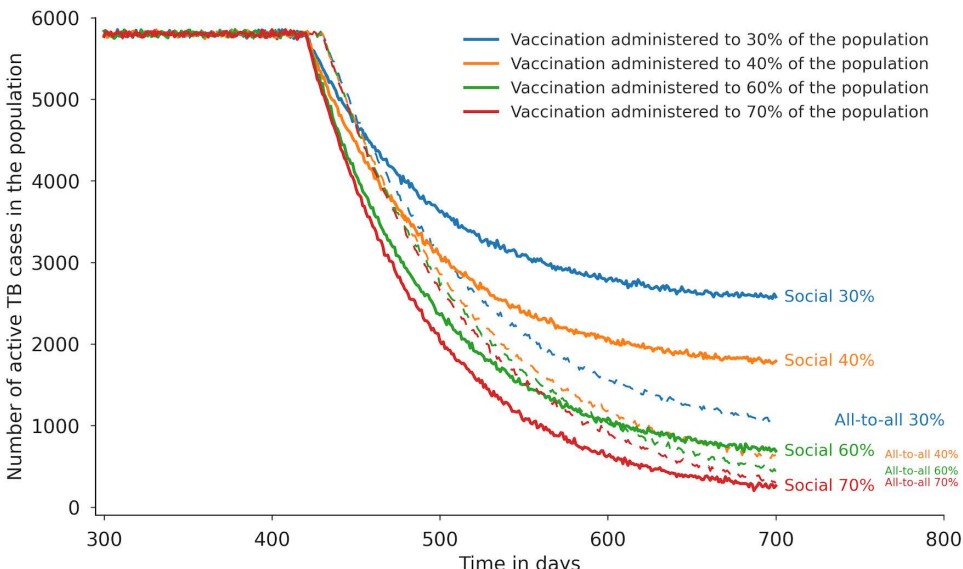

**Fig 6. Daily estimates of active TB cases from the social network model with an average of 12 contacts per person (solid lines) and the all-to-all model (dashed lines) under varying vaccination coverage levels.** Each curve shows active TB trajectories over time after vaccination with 100% efficacy administered to 30%, 40%, 60%, or 70% of the population. The all-to-all model produces steep initial declines followed by stabilization at low prevalence, with higher coverage (60–70%) nearly eliminating transmission. The social network model shows sharper early declines but reaches a higher residual plateau by day 700, particularly at lower coverage levels (30–40%), where transmission persists through highly connected individuals. At higher coverage levels, trajectories from both models converge more closely, although small residual infection remains in the social network model due to hub-driven transmission.

In denser networks, the same interventions achieved smaller absolute reductions in cases because transmission was more difficult to interrupt. However, the overall patterns were preserved: higher vaccine coverage and efficacy, or larger improvements in recovery, consistently produced greater reductions than lower-intensity interventions (Table 2). Robustness checks using multiple random network seeds showed that these findings were stable and not driven by stochastic variation in network realization.

## Discussion

This study aimed to isolate the role of contact network structure in determining projected intervention impact by directly comparing all-to-all and Barabási–Albert (BA) social network models under harmonized TB assumptions. We found that a social network model predicted smaller reductions in TB cases compared with an all-to-all model configured with identical TB natural history and intervention parameters. This divergence was most pronounced for interventions with lower coverage or smaller gains in treatment recovery rates. In contrast, when interventions were highly effective and delivered at high coverage, both models produced similar outcomes. Importantly, time-series trajectories showed that the BA model simulated initially sharper declines following vaccination, but these gains plateaued earlier and at higher prevalence levels than in the all-to-all model, reflecting the persistence of transmission in highly connected network hubs.

These findings suggest that model structure substantially influences projected intervention impact, particularly for suboptimal interventions or when interpreting short-term declines. Early improvements seen in social network simulations may not sustain long-term epidemic control, highlighting the importance of accounting for network heterogeneity when setting targets or interpreting pilot intervention results. Our results support systematic testing of the robustness of TB forecasts to social network assumptions [31–33].

**Table 2. Percentage reduction of active TB cases in the BA scale-free network model at different average network degrees (12, 20, 40, and 60 contacts per person) compared with the all-to-all model. Interventions included vaccination with 100% efficacy at 30%, 40%, 60%, and 70% coverage, and treatment improvements that increased the recovery rate by 20%, 30%, 40%, and 50%. Results represent mean reductions (± standard error) at day 700 of the simulation (300 days after intervention).**

| | | Model type | | | | |
| --- | --- | --- | --- | --- | --- | --- |
| | | All-to-all | Social network (12 contacts) | Social network (20 contacts) | Social network (40 contacts) | Social network (60 contacts) |
| **Vaccination rate** | 30% | 86 ± 1.3% | 57 ± 1.8% | 58 ± 1.1% | 58 ± 1.8% | 57 ± 0.6% |
| | 40% | 94 ± 1.3% | 72 ± 1.7% | 72 ± 1.1% | 72 ± 0.7% | 72 ± 1.2% |
| | 60% | 96 ± 0.9% | 90 ± 1.2% | 91 ± 0.3% | 91 ± 0.8% | 90 ± 0.6% |
| | 70% | 99 ± 0.3% | 96 ± 1.5% | 97 ± 0.3% | 97 ± 0.8% | 98 ± 0.5% |
| **Treatment recovery rate** | 20% | 82 ± 0.9% | 71 ± 1.4% | 71 ± 0.1% | 72 ± 1.1% | 71 ± 0.5% |
| | 30% | 98 ± 0.9% | 80 ± 1.5% | 81 ± 0.1% | 81 ± 1.1% | 80 ± 0.5% |
| | 40% | 99 ± 0.1% | 87 ± 1.5% | 87 ± 0.5% | 88 ± 1.0% | 88 ± 0.7% |
| | 50% | 100 ± 0.1% | 91 ± 1.1% | 92 ± 0.2% | 92 ± 1.2% | 92 ± 0.7% |

The vaccination coverage levels and treatment improvement values examined in this analysis were not derived from the WHO End TB Strategy targets for 2035, which define reductions in TB incidence, mortality, and catastrophic costs rather than operational thresholds for vaccination or treatment scale-up [7]. Instead, the selected intervention levels – vaccination administered to 30%, 40%, 60%, and 70% of the population, and treatment improvements increasing the recovery rate by 20–50%, were chosen as exploratory ranges commonly used in transmission modeling [28]. These values allowed us to systematically assess how contact network structure modifies projected intervention impact while holding TB natural history and epidemiological assumptions constant. They should therefore be interpreted as scenario analyses designed to isolate structural effects, rather than representations of End TB target benchmarks.

Our study has several limitations, each with implications for interpretation. First, this was a model-to-model comparison without empirical validation ("ground truth"), meaning that the findings should be viewed as illustrative of how contact structure influences intervention effects rather than as direct forecasts of real-world TB dynamics. Second, the models were not tailored to specific settings or geographic regions. This limits transferability, as patterns of TB epidemic differ across low- and high-burden settings, where contact patterns, health system capacity, and intervention coverage vary widely. Third, we modeled all-form TB without differentiating between drug-sensitive and drug-resistant subtypes. In reality, the contribution of multidrug-resistant TB (MDR-TB) and extensively drug-resistant TB (XDR-TB) to overall burden is substantial in some regions [10]. Our results therefore best reflect drug-sensitive TB and may underestimate challenges posed by resistant forms.

Fourth, we used a Barabási–Albert scale-free social network model, which assumes preferential attachment and produces hubs with disproportionately many contacts. While this captures key features of human connectivity, different communities may exhibit other structures, such as small-world or community-based networks. Using only one topology may therefore not fully capture the diversity of real-world interactions. Exploring additional network types could provide a more complete understanding of how social structures influence TB transmission.

Fifth, social determinants such as poverty, overcrowding, and stigma were not incorporated into the models. These factors can strongly influence TB incidence and may outweigh biomedical interventions in shaping epidemic dynamics [34]. Sixth, we represented all-form TB without distinguishing pulmonary and extrapulmonary manifestations. While pulmonary TB drives transmission in most populations, extrapulmonary TB occurs disproportionately in children and immunocompromised individuals, particularly those with HIV [2,35]. Our results therefore apply most directly to pulmonary TB in mixed adult populations, and caution is needed when extrapolating to groups at elevated risk of extrapulmonary disease.

Seventh, we assumed identical disease progression parameters, treatment efficacy, and vaccine efficacy across both models to isolate the influence of contact structure. In real-world populations, TB epidemics would not necessarily progress identically under homogeneous mixing versus networked contact structures. This simplification was intentional, as our goal was to highlight structural differences rather than conflate them with heterogeneity in natural history or intervention effects. Finally, we assumed a vaccine with 100% efficacy to isolate the influence of contact structure. This simplification allowed direct comparison across frameworks but does not reflect real-world TB vaccines under development, which are likely to have partial efficacy, incomplete take, and variable coverage. We also modeled vaccination as an instantaneous mass campaign, whereas in reality rollouts are gradual and often staggered across populations. Thus, our results should be interpreted as a "best-case" exploration of network effects rather than a forecast of any specific vaccine product.

These limitations mean that while our results clarify how network structure modifies projected intervention effects, they do not provide setting-specific forecasts and should not be generalized to particular populations without additional calibration. Future analyses could also vary baseline prevalence assumptions, since TB burden differs widely across settings. This would allow testing whether the relative impact of network versus all-to-all structures remains consistent under different epidemic starting conditions.

## Conclusion

When social networks were incorporated into TB transmission models, the BA scale-free framework simulated smaller reductions in projected disease than the all-to-all model, underscoring that models ignoring network heterogeneity may overestimate the long-term impact of suboptimal interventions. This direct comparison under identical TB assumptions is novel in the TB modeling literature, as it isolates the effect of contact structure itself from other factors. By clarifying this structural influence, our study provides a foundation for future extensions that incorporate drug-resistant TB, social determinants of transmission, and alternative network structures, as well as stratifying outcomes for vulnerable subgroups such as children and immunocompromised individuals. Additional priorities include calibrating to empirical epidemiological data, modeling gradual intervention rollouts, and exploring partial efficacy vaccines. Sensitivity analyses incorporating social networks should become a routine part of TB modeling to improve the robustness and applicability of epidemic forecasts.

## Acknowledgments

Thanks to Paul Kaefer and David Kaftan for code review and support.

## Author contributions

**Conceptualization:** Masabho P. Milali, Anna Bershteyn.

**Data curation:** Masabho P. Milali.

**Formal analysis:** Masabho P. Milali.

**Investigation:** Masabho P. Milali.

**Methodology:** Masabho P. Milali, Anna Bershteyn.

**Project administration:** Masabho P. Milali.

**Resources:** Masabho P. Milali.

**Software:** Masabho P. Milali, George F Corliss.

**Validation:** Masabho P. Milali, Hae-Young Kim, George F Corliss, Anna Bershteyn.

**Visualization:** Masabho P. Milali, George F Corliss, Anna Bershteyn.

**Writing – original draft:** Masabho P. Milali.

**Writing – review & editing:** Masabho P. Milali, Hae-Young Kim, George F Corliss, Anna Bershteyn.

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
