## [Decision Letter · Decision Letter 0]

11 Jul 2025

PONE-D-24-57786Do social networks attenuate the population-level impact of tuberculosis interventions? A mathematical modeling studyPLOS ONE

Dear Dr. Milali,

Thank you for submitting your manuscript to PLOS ONE. After careful consideration, we feel that it has merit but does not fully meet PLOS ONE’s publication criteria as it currently stands. Therefore, we invite you to submit a revised version of the manuscript that addresses the points raised during the review process.

We look forward to receiving your revised manuscript.

Kind regards,

Mickael Essouma, M. D.

Academic Editor

PLOS ONE

Journal Requirements:

“This work was supported by the Global Health Pilot Grant Program, Section for Global Health, Department of Population Health, NYU Langone Health.”

5. Please provide a complete Data Availability Statement in the submission form, ensuring you include all necessary access information or a reason for why you are unable to make your data freely accessible. If your research concerns only data provided within your submission, please write "All data are in the manuscript and/or supporting information files" as your Data Availability Statement.

Additional Editor Comments:

I comend the the authors for conducting a mathematical modeling study to assess whether the Barabasi-Albert scale-free mathematical model performs better than the all-in-model model when it comes to forecasting tuberculosis (TB) transmission patterns and hence epidemics in a population, given the high global burden of tuberculosis. However, there are important methodological and editing issues potentially limiting the reliability, readability and impact of the manuscript.

Major comments

1. Regarding the comment of reviewer 3 about established reporting guidelines not mentioned in the manuscript, consider complying with the EPIFORGE 2020 guidelines and report that you complied with those guidelines at the beginning of the Materials and Methods section. This entails the authors need to extend the Materials and Methods section in a way that each element required in the methods section according to the EPIFORGE 2020 guidelines would be provided in an individual sub-section. The sub-section "Ethical approval" of the Materials and Methods section needs to revised into "Ethical considerations" and moved to the end of the Materials and Methods section. The ethical considerations sub-section needs to include not only information about ethical approval, but also information about whether the authors complied with ethical rules for health research, although their research did not include human or animal participants. Plus, consider uploading an EPIFORGE 2020 checklist, based on Table 1 of the article Pol;let etal. PLoS Med 18(10): e1003793. https://doi.org/10.1371/journal.pmed.1003793, as a supplemental material to showcase how you pragmatically complied with the EPIFORGE 2020 guidelines. See: Pollett S, Johansson MA, Reich NG, Brett-Major D, Del Valle SY, Venkatramanan S, et al. (2021) Recommended reporting items for epidemic forecasting and prediction research: The EPIFORGE 2020 guidelines. PLoS Med 18(10): e1003793. https://doi.org/10.1371/journal.pmed.1003793.

2. Manuscript's title. Going through the introduction, I wonder whether the title "Are social networks better than all-in-all models for forecasting tuberculosis epidemics? A mathematical modelling study" does not resonate more with the work you conducted and reported in this manuscript.

3. Abstract. Consider adjusting its content with that of the revised main manuscript.

4. Consider inserting keywords just after the abstract.

5. Introduction. I suggest revising the introduction to better inform readers why you conducted this study and provide the study purpose in a clearer way. Accordingly, you need to extend the first paragraph with more details about the definition of tuberculosis (using updated terms: see Garcia-Basteiro et al. Lancet Glob Health. 2024 May ; 12(5): e737–e739. doi:10.1016/S2214-109X(24)00083-4.), tuberculosis subtypes (based on organ involvement, drug sensitivity and disease activity), the burden of drug-sensitive and drug-resistant tuberculosis, including details about the impact of tuberculosis treatment and vaccination on the control of the TB epidemic across low and high endemicity TB settings around the world (Chihota et al. Drugs (2025) 85:127–147 https://doi.org/10.1007/s40265-024-02131-3 and DOI 10.3389/fpubh.2024.1408316). You would then go on to describe patterns of tuberculosis transmission across low and high endemicity tuberculosis areas around the world, including risk factors for transmission and how mathematical models have helped decipher the patterns of tuberculosis transmission and hence TB epidemic around the world so far. (Chihota et al. Drugs (2025) 85:127–147 https://doi.org/10.1007/s40265-024-02131-3, DOI 10.3389/fpubh.2024.1408316 and Lo et al. Epidemics 39 (2022) 100570) While doping this, you would show how all-in-all models have mainly contributed to this endeavour whereas there is a considerable knowledge gap about how and why the Babarasi-Albert scale-free social network model could better decipher the patterns of TB transmission (in which population specifically?) and hence better forecast and prevent TB epidemics in (some or all) populations. You need to do this with more details than actually but in a more simplistic way. See how Wikipedia has commented on the Barabasi-Albert scale-free social network model: https://en.wikipedia.org/wiki/Barab%C3%A1si%E2%80%93Albert_model) Following this, the authors would usher in the study purpose.

6. Materials and Methods. Along with my comment above raising the need to subdivide the Methods section into many sub-sections to improve its readability, I have other comments about this section. What TB subtypes did you incorporate in the models: all-type tuberculosis as it seems in the discussion section? Consider specifying in this section. Why did you assume that "the same underlying progression" and treatment and vaccine efficacy would be observed in both studied models? Would the TB epidemic progress in the same way in both scenarios in real life? What did you mean buy "steady-state" in the context of this study? Consider specifying. It is in the discussion section that you first specified that the population incorporated in the models was not representative of any specific setting or geographic location , but it is in this section that you need to clearly specify that and that you did not stratify the population into low and highest-risk (e.g., under-five children, older adults, immunocompromised (especially HIV/AIDS) patients, incarcerated individuals, migrants [especially refugees, asylum seekers, war/torture/crisis survivors]) and most visible community members. Along this line, the authors need to more precisely enumerate these population groups when addressing the study limitations because the TB epidemic is virtually different in these population groups than in other lower-risk population groups. Furthermore, when calling expanded modelling efforts incorporating real-world social network complexities, the authors would like to more precisely address the potential influence of most visible community members in TB transmission around the world (https://en.wikipedia.org/wiki/Barab%C3%A1si%E2%80%93Albert_model), given that in the current context of globalization, most people encounter Mycobacterium tuberculosis at some point of their life and keep it in their bodies throughout their lifetime. See Chihota et al. Drugs (2025) 85:127–147 and current reference 29. The age group of the represented population was not mentioned in this section. There is a need for a sub-section on definitions of terms (e.g., incidence, prevalence, treatment recovery/efficacy, treatment adherence, imperfect adherence, treatment efficacy, vaccine efficacy and vaccine coverage) in this section given the strong comments you made in the discussion section using some of those terms. What treatments did you consider in the models? Same question for vaccines knowing that BCG remains the only certified TB vaccine to the best of my knowledge. See https://doi.org/10.1371/journal.pone.0317233. It should also be clear that the BCG is mainly recommended in young populations owing to its limited efficacy in adults, and BCG mass vaccination is uncommon in low endemicity areas whereas the authors claim that the model did not consider any geographic boundary. Can you ponder about this in this section and in the limitations statement of the discussion section? What guided the choice of treatment recovery rate (20, 30, 40, 50 percent), vaccine coverage rate (40, 50, 60, 70) and time schedule thresholds (400 days to reach the equilibrium, 300 days after the start of the treatment (or 700 days after launching the model) to estimate the impact of interventions? What did you mean by "model reaching an equilibrium"? Why did you need the equilibrium to be reached to simulate the impact of TB interventions? Could you elaborate more on the procesd of simulating the impact of interventions? All these precision's are warranted. You performed sensitivity analyses, but you also need to provide information on results calibration and validation, at least in the limitations statement of the discussion section. See Lo et al. Epidemics 39 (2022) 100570).

7. Results. Could you separate the presentation of results of the all-in-all model from that of the Barabasi-Albert scale-free social network model? Consider further adjudicating the results in agreement with revisions of the Materials and Methods section.

8. Discussion and conclusion sections. I appreciate the enumeration of study limitations. However, the authors should consider specifying the impact of each major limitation on the interpretation of satudy results. Fpor example, when stating that "our study did not focus on specific settings or geographical regions.", they would better complete the statement by stating something like:@ "because patterns of TB epidemic vary across low and and high endemicity TB settings." Also, besides the limitations highlighted in the current manuscript, the authors need to specify whether this study only considered drug-sensitive tuberculosis as in previous with tuberculosis modelling studies, in a context where the relative contribution of drug-sensitive and drug-resistant tuberculosis to the overall burden of tuberculosis in real-life may not be well known (Song HW, Tian JH, Song HP, Guo SJ, Lin YH and Pan JS (2024) Tracking multidrug resistant tuberculosis: a 30-year analysis of global, regional, and national trends. Front. Public Health 12:1408316. doi: 10.3389/fpubh.2024.1408316). The non incorporation of social determinants (notably in the social network) was also a major limitation given that their influence on population incidence of tuberculosis may be more important than that of TB treatment. See lange et al. Pathog Immun. 2025 Mar 2;10(2):1-45. doi: 10.20411/pai.v10i2.791. At the end of this manuscript, the population group in which this study would best help prevent the spread of the Tb epidemic is unclear. Likewise, the TB subtype (at least based on organ involvement pulmonary and extrapulmonary TB) this study would best help prevent is unclear. Even if the authors considered all-form tuberculosis, the risk for incident extrapulmonary TB is more common in under-five children and immunocompromised persons than in other population groups whereas pulmonary tuberculosis (at least pulmonary tuberculosis infection [Garcia-Basteiro eyt al. https://doi.org/10.1016/s2214-109x(24)00083-4]) might be more evenly distributed among immunocompetent and immunocompromised persons. See https://doi.org/10.3390/jor1020015 and Guzmán-Beltrán, S.; Bobadilla, K.; Santos-Mendoza, T.; Flores-Valdez, M.A.; Gutiérrez-González, L.H.; González, Y. Human Pulmonary Tuberculosis: Understanding the Immune Response in the Bronchoalveolar System. Biomolecules 2022, 12, 1148. https://doi.org/10.3390/biom12081148. These issues need to be addressed in the discussion and conclusion sections.

Minor comments

1. Consider formatting the manuscript as recommended in PLOS One author guidelines, including the references.

2. Some language editing is warranted. For instance, I advise against using the words "predict" and "forecast" in the results section, but rather providing results of model simulation. Then, in the discussion section, you would make statements like: "Results of the all-in-all model would allow the prediction or forecasting of ....in a scenario where an infected individual may transmit TB to any individual of the same population, whereas results of the Barabasi-Albert scale-free social network would allow the prediction of .... in a scenario where an infected individual would transmit TB only to their social contacts.".

3. References need to be updated and strengthened.

4. Consider numbering the manuscript's lines and sending figures and tables at the end of the manuscript to ease the assessment of the next version of the manuscript.

5. Where are statements about conflicts of interests and data availability that should be found at the end of the manuscript before the reference section?

6. Where in the manuscript did you state that there is a preprint for this manuscript (https://doi.org/10.21203/rs.3.rs-3283210/v1)?

Mickael ESSOUMA

Reviewers' comments:

Reviewer's Responses to Questions

**Comments to the Author**

1. Is the manuscript technically sound, and do the data support the conclusions?

Reviewer #1: Yes

Reviewer #2: Yes

Reviewer #3: Partly

Reviewer #4: Partly

Reviewer #5: Yes

Reviewer #6: Partly

2. Has the statistical analysis been performed appropriately and rigorously?

Reviewer #1: Yes

Reviewer #2: Yes

Reviewer #3: I Don't Know

Reviewer #4: No

Reviewer #5: Yes

Reviewer #6: No

3. Have the authors made all data underlying the findings in their manuscript fully available?

Reviewer #1: Yes

Reviewer #2: Yes

Reviewer #3: No

Reviewer #4: Yes

Reviewer #5: Yes

Reviewer #6: Yes

4. Is the manuscript presented in an intelligible fashion and written in standard English?

Reviewer #1: Yes

Reviewer #2: Yes

Reviewer #3: Yes

Reviewer #4: Yes

Reviewer #5: Yes

Reviewer #6: No

5. Review Comments to the Author

Reviewer #1: •The study presents the results of original research.

The paper highlights social network attenuates the population-level impact of tuberculosis interventions and the study is demonstrated using mathematical model.

The work can be published in the journal PLOS ONE provided the following issues can be addressed.

1.System of ordinary differential equations is presented as partial differential equations. The notation of equations (1-5) need to be corrected.

2.The authors can check paper [29] on how system of ordinary differential equation is formulated and presented.

3.The schematic diagrams (A /B) are shows as figures under fig 2. The schematic diagrams must be shown before formulating the system of ordinary differential equations (ODE’s)

4.The figures are shown in different pages without caption. The captions are presented elsewhere. It is very difficult to follow the sequence of this paper.

•Results reported have not been published elsewhere.

The results have not been published elsewhere, the authors need to put this ideas in a way which readability will be easy.

•Experiments, statistics, and other analyses are performed to a high technical standard and are described in sufficient detail.

The statistical results are presented very well and they were generated using R software.

•Conclusions are presented in an appropriate fashion and are supported by the data.

The conclusion is presented in the correct scientific format and it validate the presented mathematical models.

•The article is presented in an intelligible fashion and is written in standard English.

Yes, the article is written and presented in the correct English language.

•The research meets all applicable standards for the ethics of experimentation and research integrity.

Yes

•The article adheres to appropriate reporting guidelines and community standards for data availability.

I am comfortable with the report, it adheres to all the guidelines.

Final outcome: The paper can be published provided all the recommended corrections are adhered to.

Your Truly

Dr MC Kekana (12 June 2025)

Reviewer #2: This study investigates whether social networks diminish the projected impact of tuberculosis (TB) interventions using mathematical modeling. Researchers developed two TB transmission models with identical assumptions—one assuming "all-to-all" mixing (where every individual can infect any other) and the other using a Barabási-Albert scale-free social network (where infection occurs only among social contacts).

Interventions including vaccination and treatment were simulated across varying coverage and efficacy levels. The social network model consistently projected lower reductions in TB incidence, especially under low-coverage or low-efficacy scenarios. For instance, a vaccine with 30% coverage and full efficacy reduced cases by 72% in the network model versus 94% in the all-to-all model. Similarly, modest treatment efficacy improvements yielded significantly smaller impacts in the network-based approach. These differences persisted across varying population sizes and network densities. The study concludes that traditional all-to-all models may overestimate the benefits of sub-optimal interventions and thus influence unrealistic target-setting in TB control strategies.

Although high-efficacy and high-coverage interventions showed consistent impact across both models, the inclusion of network structures is crucial in forecasting TB trends more accurately. Limitations include generalized assumptions, single network type, and simplifications in vaccine modeling. The authors call for expanded modeling efforts incorporating real-world social network complexities.

I find that this study is of real interest and has the merit to be considered for publication. Its primary strength lies in its rigorous head-to-head comparison of two modeling paradigms under consistent disease dynamics and intervention parameters. The use of a Barabási-Albert scale-free network brings realism to contact patterns and highlights the role of heterogeneity in TB transmission. It challenges common assumptions in TB modeling and stresses caution in policy planning and its findings are robust across multiple scenarios and offer valuable insights into the implications of model structure on forecasting and public health decision-making.

However, I also note that there are several issues that may need to be addressed for further improvement. I recommend that the authors revise their manuscript while addressing each of the following points:

•The model is not tailored to specific geographic or demographic settings. Can the authors relate their work to real world epidemiological data?

•The authors should include in the discussions section a few lines about the possibility of exploring additional network topologies (e.g., small-world or community-based) to explain how structural differences influence intervention outcomes.

•Instantaneous vaccination is an issue of concern in the model. Real life scenarios would have gradual roll-out of vaccination. I suggest to address this issue as well in the discussions.

•The issue of vaccine availability as well as timing of introduction is discussed and modelled in the literature. One such study that the authors can refer to on their introduction or in their discussion while addressing the previous points is the following: https://doi.org/10.1371/journal.pone.0267840

•Assumption of 100% vaccine efficacy is very ideal and usually not true. I suggest that in the model results,the authors consider a bracket of efficacies, for example 80-85-90-95% scenarios, with corresponding plots and comparisons. This could be done 1 figure, then move to a particular benchmark in the rest of the results, maybe 90% per se.

•Social determinants affecting treatment uptake and exposure risk were not considered. Explain how such factors could improve projections and guide equity-oriented interventions.

I recommend a major revision to specifically address these points, and, if treated well, the paper would be suitable for consideration afterwards.

Reviewer #3: This study examines an important question about tuberculosis modeling by comparing traditional "all-to-all" mixing models with social network models to understand how different modeling approaches might affect projected intervention impacts. This is a relevant topic given the ongoing challenges in meeting WHO TB elimination targets. The research design appears sound for comparing modeling approaches, and the findings about different impacts in low-coverage versus high-coverage scenarios could be valuable for the TB modeling community.

I have identified several issues that need to be addressed to meet PLOS ONE publication standards:

Major issues:

1.The manuscript lacks mandatory elements for PLOS ONE publication. For example, there is no data availability statement indicating where model code, parameter values, and analysis scripts will be made available. There is also no reference to established reporting guidelines for mathematical modeling studies. The ethics statement, while appropriate for a modeling study, should be expanded to specifically mention data sources used and confirm no human subjects were involved.

2.It appears the author might have made some claims that extend well beyond what the modeling results can support, in the discussion section. The claims include:

a.Statement that findings "call into question the sufficiency of resources currently allocated to global TB programs"

b.Assertion that "even greater resources may be necessary"

c.Call for "Redoubled efforts are needed to invest in more effective and accessible vaccination..."

These statements read more like policy advocacy than scientific conclusions. The manuscript would be strengthened if the authors instead focused on the specific modeling implications with quantified results. For example, instead of making broad resource allocation claims, the authors could indicate that their findings suggest social network models reduce projected efficacy of sub-optimal interventions by X–Y percentage points, potentially affecting the feasibility of current targets when interventions have limited coverage

3.Although important limitations were acknowledged, the authors did not explain how the limitations affected the reliability of their study conclusions (including policy recommendations) or connect the limitations to the validity of their study findings.

Minor issues:

4.There are language and clarity problems throughout the manuscript:

a.Several sentences are unnecessarily complex and should be simplified (an example is the second sentence in the abstract (“We compared the impact of TB treatment and vaccination in an all-to-all compartmental model versus a social network model that…). This sentence tries to pack too much information into one complex statement with multiple clauses, making it harder to follow

b.Inconsistent formatting of "all-to-all" (sometimes in quotes, sometimes not)

c.Ambiguous phrasing like "plateau relative to the 'all-to-all' model". The authors should clarify whether the incidence plateaued at higher levels in the network model.

d.Informal language like "Redoubled efforts" should be replaced with more precise academic language

5.The discussion should begin with a summary of what the research achieved relative to the authors’ stated aims, rather than immediately diving into broad implications.

6.The conclusion lacks precision in the use of implied statements. For example the phrase "large implications" needs quantification. It would be better if he authors could specify the actual magnitude of divergence they observed. This would give readers concrete understanding of when these modeling differences matter most.

Reviewer #4: I would like to thank the authors for their manuscript and for continuing to endeavour in the application of computational techniques to TB control and prevention. TB remains a significant challenge, especially in low resource settings and as the authors indicated, mathematical modelling is an important tool to investigate many aspects safely and cost-effectively.

Methods: Model development

I would suggest the authors review the model equations and diagram:

1. The model diagram and equations specify that those persons that leave each compartment due to death (either natural or disease related) re-enter the susceptible compartment. This cannot be, since persons that leave compartments due to death leave the system entirely, despite the birth and death rates being equal.

2. The inflow into the S compartment and outflow from S to V in diagram B is not reasonable. Persons are recruited or born into the S compartment at a rate (the birth/recruitment rate). From this compartment a proportion can be vaccinated (lambda) at the vaccination rate and enter the V compartment. The first aspect I suggest the authors revisit is the inflow into the S compartment. The (1-lambda) does not fit since you are saying that those of the S compartment that were not vaccinated are recruited into S. All persons enter S unvaccinated. Thereafter they either proceed to V or the remaining unvaccinated proceed along to E or die out of the S compartment. The second aspect is the movement from S to V. The vaccination rate is indicated to be the death/birth rate. Is this a valid assumption? The third aspect is the movement of the remaining S population to E and further. Not all persons proceed along that route, but only those that were not vaccinated, i.e., the proportion (1-lambda).

A further question is whether the authors assume the vaccinated individuals incur life-long immunity or whether they loose protection at some point and re-enter the S compartment?

3. An important assumption not provided within the model description is that it is assumed that disease related death only occurs among those with active TB, i.e., in the I compartment. Further, despite disease related death from I, do you further assume that those with active TB do not undergo natural death?

4. The model equations are missing some terms, namely in equation 1 and 2, the vaccination rate is not included in the equation. Equation 3, (1-p)*sigma*E is missing.

5. It would be of benefit for the readers -especially those without an infectious disease modelling background - that you provide some explanation of a scale-free network model - particularly what the major assumptions are- and also clearly outline your process of calibration. Further, although the link to the code repository is provided, it would be beneficial for the readers to get a brief overview of what the code actually does, generating the figures provided.

6. Model equations are formulated as PDEs pointing towards equations that are functions of more than one variable. Time is clearly specified, but what is the other? The authors further specify the use of 'deSolve' which primarily works with ODEs without some creative extension. Should the latter be the case, please include a clear description of the methodology behind the extension.

7. The methods would benefit from a clear description of the model (as outlined in the model diagram) including the assumptions defining the transitions and parameter selections.

Minor errors:

1. Under Model development, line 3: "emplying" should be "employing"

Reviewer #5: Dear Editor,

The authors presented novel study of “Do social networks attenuate the population-level impact of tuberculosis interventions? A mathematical modeling study”. The manuscript is reviewed by keeping in view its suitability, scope of the journal, and originality. I recommend this manuscript for publication in revised form. The author need to provide point-by-point response or rebuttal to my observations listed below.

•Include some major outcomes in the abstract.

•To extend this idea to the utilization of nanotechnology in the assumed model consult following works:

http://dx.doi.org/10.1016/j.rinp.2024.107863, http://dx.doi.org/10.1016/j.ijheatfluidflow.2024.109322, http://dx.doi.org/10.1038/s41598-023-48364-2, https://doi.org/10.1016/j.csite.2024.104178, http://dx.doi.org/10.1177/14613484231216198, http://dx.doi.org/10.1016/j.asej.2023.102316

•Does the manuscript address an important gap in tuberculosis modeling by comparing "all-to-all" and social network-based transmission structures?

•How novel is the direct comparison between these two modeling frameworks, particularly when using identical TB assumptions?

•Could the authors better explain how this work differs from previous modeling studies of TB or other infectious diseases?

•The study assumes 100% vaccine efficacy in many simulations. Can the authors justify this assumption in the context of current vaccine development?

•Have the authors considered including scenarios with more realistic vaccine efficacies (e.g., 50–70%) or “take” rates?

•How were the transmission parameters (β) calibrated to ensure that both models produce similar baseline TB prevalence?

•What justifies the specific prevalence level used for calibration?

•Would sensitivity analysis around baseline prevalence assumptions add robustness to the findings?

•The conclusions follow from the presented results. I suggest that it is betters is authors can provide the future research prospective at the end of article.

Reviewer #6: The manuscript tackles an important and timely topic in tuberculosis (TB) modeling by contrasting the conventional all-to-all mixing assumption with a more realistic social network-based transmission framework. While the overall premise is relevant and the comparative approach has potential value for informing TB control strategies, the current version of the manuscript suffers from several conceptual and methodological shortcomings that affect its clarity and impact. In particular, there are concerns related to key model assumptions and their biological plausibility. My specific concerns and suggestions are outlined below:

1. A more detailed explanation of the model is necessary to make it accessible to readers from various disciplines. The numerical simulation is presented over a span of more than 700 days, which cannot be considered a particularly short time frame. Is it essential to assume that the total population remains constant over this long time span? The current model appears unnecessarily complex primarily to preserve this assumption. There are much simpler and more straightforward ways to maintain a constant population without introducing such complications. It seems challenging—perhaps even impossible—to provide a detailed and biologically sound explanation of the current version of the transfer diagram and, by extension, the model itself. For example, the biological meaning of the last term in model equation (1) is unclear in the context of TB transmission.

2. Moreover, some parts of the model equations do not match the transitions shown in the transfer diagram in Fig. 2(b).

3. In real-world settings, vaccinated individuals are classified in a separate group, and due to imperfect vaccination, some may still become infected. The model should reflect that breakthrough infections happen directly from the vaccinated compartment, not by moving first to the susceptible compartment. In practice, we only realize the vaccine’s imperfection once vaccinated individuals become infected. Therefore, it is more appropriate for the infection to occur from the vaccinated group (V), rather than first moving them to S and then to E/I.

4. It is unclear how the two models were calibrated to produce similar dynamics despite using different values of the transmission rate β (beta)?

5. The simulation technique for the network model also requires further elaboration. A clearer description is needed to help readers understand how and why the compartmental model falls short compared to the network-based implementation.

The true potential of the manuscript can only be fully assessed once the identified issues are adequately addressed. I recommend that the authors revise the manuscript by resolving these concerns, providing a comprehensive and transparent description of the model and methods, and consider resubmitting the work as a new submission for further evaluation

6. PLOS authors have the option to publish the peer review history of their article (what does this mean?). If published, this will include your full peer review and any attached files.

Reviewer #1: No

Reviewer #2: No

Reviewer #3: **Yes:** Eric Osayemwenre Iyahen

Reviewer #4: No

Reviewer #5: **Yes:** Muhammad Abdul Basit

Reviewer #6: No

---

## [Author Response · Author response to Decision Letter 1]

5 Nov 2025

September 3, 2025

Re: Resubmission of Manuscript PONE-D-24-57786, Do social networks attenuate the population-level impact of tuberculosis interventions? A mathematical modeling study.

Dear Dr. Mickael Essouma and Reviewers,

We would like to thank you for the constructive and insightful feedback on our manuscript. Below we provide a point-by-point response. Reviewer’s comments are reproduced in italics, and our responses follow in underlined regular text.

We hope that our revisions have addressed all comments and improved the manuscript. We thank the reviewers and editor once again for their thoughtful feedback.

Sincerely,

Masabho P. Milali, PhD

Editor comments

https://journals.plos.org/plosone/s/file?id=wjVg/PLOSOne_formatting_sample_main_body.pdf and https://journals.plos.org/plosone/s/file? id=ba62/PLOSOne_formatting_sample_title_authors_affiliations.pdf

Response: We now confirm that the resubmission was prepared using the official PLOS ONE style templates with corrected file naming conventions.

Response: We now remove all funding mentions from the main text and retain funding details only in the Funding Statement. (Lines 405 – 408)

Response: We now reconcile Funding and Financial Disclosure details. (Lines 405-408)

“This work was supported by the Global Health Pilot Grant Program, Section for Global Health, Department of Population Health, NYU Langone Health.”Please state what role the funders took in the study. If the funders had no role, please state: "The funders had no role in study design, data collection and analysis, decision to publish, or preparation of the manuscript." If this statement is not correct you must amend it as needed. Please include this amended Role of Funder statement in your cover letter; we will change the online submission form on your behalf.

Response: We now add: “The funders had no role in study design, data collection and analysis, decision to publish, or preparation of the manuscript”. (Lines 405-408).

5. Please provide a complete Data Availability Statement in the submission form, ensuring you include all necessary access information or a reason for why you are unable to make your data freely accessible. If your research concerns only data provided within your submission, please write "All data are in the manuscript and/or supporting information files" as your Data Availability Statement.

Response: We now state: “All data are in the manuscript”. (Lines 257-260, 418-421)

Additional Editor Comments:

I comend the the authors for conducting a mathematical modeling study to assess whether the Barabasi-Albert scale-free mathematical model performs better than the all-in-model model when it comes to forecasting tuberculosis (TB) transmission patterns and hence epidemics in a population, given the high global burden of tuberculosis.

Response: We thank the editor for the positive feedback

However, there are important methodological and editing issues potentially limiting the reliability, readability and impact of the manuscript.

Major comments

1. Regarding the comment of reviewer 3 about established reporting guidelines not mentioned in the manuscript, consider complying with the EPIFORGE 2020 guidelines and report that you complied with those guidelines at the beginning of the Materials and Methods section. This entails the authors need to extend the Materials and Methods section in a way that each element required in the methods section according to the EPIFORGE 2020 guidelines would be provided in an individual sub-section. The sub-section "Ethical approval" of the Materials and Methods section needs to revised into "Ethical considerations" and moved to the end of the Materials and Methods section. The ethical considerations sub-section needs to include not only information about ethical approval, but also information about whether the authors complied with ethical rules for health research, although their research did not include human or animal participants. Plus, consider uploading an EPIFORGE 2020 checklist, based on Table 1 of

the article Pol;let etal. PLoS Med 18(10): e1003793. https://doi.org/10.1371/journal.pmed.1003793, as a supplemental material to showcase how you pragmatically complied with the EPIFORGE 2020 guidelines. See: Pollett S, Johansson MA, Reich NG, Brett-Major D, Del Valle SY, Venkatramanan S, et al. (2021) Recommended reporting items for epidemic forecasting and prediction research: The EPIFORGE 2020 guidelines. PLoS Med 18(10): e1003793. https://doi.org/10.1371/journal.pmed.1003793.

Response: We thank the editor for this important recommendation. The revised manuscript now complies with the EPIFORGE 2020 guidelines. At the beginning of the Materials and Methods section, we explicitly state our adherence to EPIFORGE, and the section has been restructured into individual sub-sections following the guideline items. The former “Ethical approval” subsection has been revised to “Ethical considerations” and moved to the end of the Materials and Methods. This subsection now specifies compliance with ethical standards for modeling research, even though no human or animal participants were involved. An EPIFORGE 2020 checklist, based on Table 1 of Pollett et al. (2021), has also been prepared and uploaded as supplemental material to document our compliance. Changes reflected in lines 92 - 260

2. Manuscript's title. Going through the introduction, I wonder whether the title "Are social networks better than all-in-all models for forecasting tuberculosis epidemics? A mathematical modelling study" does not resonate more with the work you conducted and reported in this manuscript.

Response: While we agree with your proposed framing, we revised the title to remove the phrase “better than” to avoid implying superiority between models, since the study was not fit to empirical data and therefore cannot establish whether one model is “better.” Instead, our comparison highlights how different assumptions about contact structure influence projections. The revised title is:

“How do social network models compare to all-to-all models for forecasting tuberculosis epidemics? A mathematical modeling study.”

We believe this phrasing captures the comparative nature you recommended while avoiding unintended claims of superiority, and ensures the title remains scientifically accurate, testable, and aligned with the scope of our study (Lines 1–2)

3. Abstract. Consider adjusting its content with that of the revised main manuscript.

Response: Abstract rewritten to match updated sections of the manuscript. (Lines 12-26)

4. Consider inserting keywords just after the abstract.

Response: Keywords inserted just after the abstract. (Lines 29-30)

5. Introduction. I suggest revising the introduction to better inform readers why you conducted this study and provide the study purpose in a clearer way. Accordingly, you need to extend the first paragraph with more details about the definition of tuberculosis (using updated terms: see Garcia-Basteiro et al. Lancet Glob Health. 2024 May ; 12(5): e737–e739. doi:10.1016/S2214- 109X(24)00083-4.), tuberculosis subtypes (based on organ involvement, drug sensitivity and disease activity), the burden of drug-sensitive and drug-resistant tuberculosis, including details about the impact of tuberculosis treatment and vaccination on the control of the TB epidemic across low and high endemicity TB settings around the world (Chihota et al. Drugs (2025) 85:127–147 https://doi.org/10.1007/s40265-024-02131-3 and DOI 10.3389/fpubh.2024.1408316). You would then go on to describe patterns of tuberculosis transmission across low and high endemicity tuberculosis areas around the world, including risk factors for transmission and how mathematical models have helped decipher the patterns of

tuberculosis transmission and hence TB epidemic around the world so far. (Chihota et al. Drugs (2025) 85:127–147 https://doi.org/10.1007/s40265-024-02131-3, DOI 10.3389/fpubh.2024.1408316 and Lo et al. Epidemics 39 (2022) 100570) While doping this, you would show how all-in-all models have mainly contributed to this endeavour whereas there is a considerable knowledge gap about how and why the Babarasi-Albert scale-free social network model could better decipher the patterns of TB transmission (in which population specifically?) and hence better forecast and prevent TB epidemics in (some or all) populations. You need to do this with more details than actually but in a more simplistic way. See how Wikipedia has commented on the Barabasi-Albert scale-free social network model: https://en.wikipedia.org/wiki/Barab%C3%A1si%E2%80%93Albert_model) Following this, the authors would usher in the study purpose.

Response: The Introduction has been expanded to incorporate the requested elements, including updated definitions of tuberculosis (Garcia-Basteiro et al., 2024), tuberculosis subtypes, and the burden of drug-sensitive and drug-resistant TB. We have also added details on the impact of treatment and vaccination across high- and low-burden settings, as well as on patterns of TB transmission and the contribution of mathematical models to understanding epidemic dynamics (Chihota et al., 2025; Lo et al., 2022). The knowledge gap regarding the Barabási–Albert scale-free model is now highlighted, and the study’s purpose has been clarified. (Lines 32–64)

6. Materials and Methods. Along with my comment above raising the need to subdivide the Methods section into many sub-sections to improve its readability, I have other comments about this section. What TB subtypes did you incorporate in the models: all-type tuberculosis as it seems in the discussion section? Consider specifying in this section. Why did you assume that "the same underlying progression" and treatment and vaccine efficacy would be observed in both studied models? Would the TB epidemic progress in the same way in both scenarios in real life? What did you mean buy "steady-state" in the context of this study? Consider specifying. It is in the discussion section that you first specified that the population incorporated in the models was not representative of any specific setting or geographic location , but it is in this section that you need to clearly specify that and that you did not stratify the population into low and highest-risk (e.g., under-five children, older adults, immunocompromised (especially HIV/AIDS) patients, incarcerated individuals, migrants [especially refugees, asylum seekers, war/torture/crisis survivors]) and most visible community members. Along this line, the authors need to more precisely enumerate these population groups when addressing the study limitations because the TB epidemic is virtually different in these population groups than in other lower-risk population groups. Furthermore, when calling expanded modelling efforts incorporating real-world social network complexities, the authors would like to more precisely address the potential influence of most visible community members in TB transmission around the world (https://en.wikipedia.org/wiki/Barab%C3%A1si%E2%80%93Albert_model), given that in the current context of globalization, most people encounter Mycobacterium tuberculosis at some point of their life and keep it in their bodies throughout their lifetime. See Chihota et al. Drugs (2025) 85:127–147 and current reference 29. The age group of the represented population was not mentioned in this section. There is a need for a sub-section on definitions of terms (e.g., incidence, prevalence, treatment recovery/efficacy, treatment adherence, imperfect adherence, treatment efficacy, vaccine efficacy and vaccine coverage) in this section given the strong comments you made in the discussion section using some of those terms. What treatments did you consider in the models? Same question for vaccines knowing that BCG remains the only certified TB vaccine to the best of my knowledge. See https://doi.org/10.1371/journal.pone.0317233. It should also be clear that the BCG is mainly recommended in young populations owing to its limited efficacy in adults, and BCG mass vaccination is uncommon in low endemicity areas whereas the authors claim that the model did not consider any geographic boundary. Can you ponder about this in this section and in the limitations statement of the discussion section? What guided the choice of treatment recovery rate (20, 30, 40, 50 percent), vaccine coverage rate (40, 50, 60, 70) and time schedule thresholds (400 days to reach the equilibrium, 300 days after the start of the treatment (or 700 days after launching the model) to estimate the impact of interventions? What did you mean by "model reaching an equilibrium"? Why did you need the equilibrium to be reached to simulate the impact of TB interventions? Could you elaborate more on the procesd of simulating the impact of interventions? All these precision's are warranted. You performed sensitivity analyses, but you also need to provide information on results calibration and validation, at least in the limitations statement of the discussion section. See Lo et al. Epidemics 39 (2022) 100570).

Response: We carefully revised the Materials and Methods to address each point raised.

• Along with my comment above raising the need to subdivide the Methods section into many sub-sections to improve its readability

Response: We now explicitly state compliance with the EPIFORGE 2020 guidelines at the start of the Methods (line 93). The section has been restructured into EPIFORGE-compliant subsections: Study Overview, Model Structure and Assumptions, Definitions of Terms, Parameterization and Calibration, Intervention Scenarios, Outcome Measures, Analysis Methods, Ethical Considerations, and Reproducibility (line 92–260).

• What TB subtypes did you incorporate in the models: all-type tuberculosis as it seems in the discussion section?

Response: We now clarify that the models represent all-form TB without stratification by organ involvement or drug resistance. This is stated explicitly in the Study Overview (line 109–111)

• Why did you assume that "the same underlying progression" and treatment and vaccine efficacy would be observed in both studied models? Would the TB epidemic progress in the same way in both scenarios in real life?

Response: We now explain that assuming identical disease progression, treatment efficacy, and vaccine efficacy across both models is a deliberate simplification to isolate the effect of contact structure. We note that in real-world scenarios, TB epidemics may not progress identically under homogeneous versus networked contact structures, but this difference is precisely the focus of our study. This point is reflected in the revised Methods (lines 201–205) and reinforced in the Discussion (lines 374–379).

• What did you mean by "steady-state" in the context of this study?

Response: We now define steady-state as “a stable epidemiological condition in which TB incidence and prevalence remain constant over time in the absence of new interventions” (Definitions of Terms, lines 184–186).

• It is in the discussion section that you first specified that the population incorporated in the models was not representative of any specific setting or geographic location , but it is in this section that you need to clearly specify that and that you did not stratify the population into low and highest-risk (e.g., under-five children, older adults, immunocompromised (especially HIV/AIDS) patients, incarcerated indiv

---

## [Decision Letter · Decision Letter 1]

18 Nov 2025

PONE-D-24-57786R1How do social network models compare to all-to-all models for forecasting tuberculosis epidemics? A mathematical modeling studyPLOS ONE

Dear Dr. Milali,

Thank you for submitting your manuscript to PLOS ONE. After careful consideration, we feel that it has merit but does not fully meet PLOS ONE’s publication criteria as it currently stands. Therefore, we invite you to submit a revised version of the manuscript that addresses the points raised during the review process.

We look forward to receiving your revised manuscript.

Kind regards,

Mickael Essouma, M. D.

Academic Editor

PLOS ONE

Journal Requirements:

Additional Editor Comments (if provided):

The authors have well addressed most of my comments. However, regarding the EPIFORGE guidelines and to avoid redundancy, details in lines 186-191 and in lines 200-205 need to be moved to the end of the introduction where the authors state the aim of the study. There is a need to revise the presentation of figures in this manuscript. For instance, the footnote of Fig 1 describes panels A and B, but I do not see those panels on Fig 1. Line 36: is it Fig 1 or Fig 2? What are Figures 4 and 6? Fig. 3, 4, 5 and 6 should be clearly specified each (including any intrinsic panels) with its footnote and title together as you did for Fig 1 or Fig 2. Could you specify the published literature you are mentioning in line 293? Did the End TB by 2035 strategy inform the target figures for treatment and vaccination used in this study or they were set arbitrarily? This should be addressed in the discussion as well.

Mickael Essouma, M.D.

Reviewers' comments:

Reviewer's Responses to Questions

**Comments to the Author**

1. If the authors have adequately addressed your comments raised in a previous round of review and you feel that this manuscript is now acceptable for publication, you may indicate that here to bypass the “Comments to the Author” section, enter your conflict of interest statement in the “Confidential to Editor” section, and submit your "Accept" recommendation.

Reviewer #2: (No Response)

Reviewer #3: All comments have been addressed

Reviewer #4: All comments have been addressed

Reviewer #5: (No Response)

Reviewer #6: All comments have been addressed

2. Is the manuscript technically sound, and do the data support the conclusions?

Reviewer #2: Yes

Reviewer #3: Partly

Reviewer #4: Yes

Reviewer #5: Partly

Reviewer #6: Yes

3. Has the statistical analysis been performed appropriately and rigorously?

Reviewer #2: (No Response)

Reviewer #3: I Don't Know

Reviewer #4: Yes

Reviewer #5: N/A

Reviewer #6: Yes

4. Have the authors made all data underlying the findings in their manuscript fully available?

Reviewer #2: Yes

Reviewer #3: Yes

Reviewer #4: Yes

Reviewer #5: Yes

Reviewer #6: Yes

5. Is the manuscript presented in an intelligible fashion and written in standard English?

Reviewer #2: Yes

Reviewer #3: Yes

Reviewer #4: Yes

Reviewer #5: No

Reviewer #6: Yes

6. Review Comments to the Author

Reviewer #2: (No Response)

Reviewer #3: (No Response)

Reviewer #4: I thank the authors for their careful consideration of the recommendations and congratulate them on an excellent piece of research.

Reviewer #5: There are some critical points those are need to be addressed before acceptance, I reserve my decision till then.

Reviewer #6: The authors have adequately addressed my concerns, and the manuscript can now proceed to publication.

7. PLOS authors have the option to publish the peer review history of their article (what does this mean?). If published, this will include your full peer review and any attached files.

Reviewer #2: No

Reviewer #3: **Yes:** Eric Osayemwenre Iyahen

Reviewer #4: No

Reviewer #5: **Yes:** Dr. Muhammad Abdul Basit

Reviewer #6: No

---

## [Author Response · Author response to Decision Letter 2]

2 Feb 2026

February 2, 2025

Re: Second Resubmission of Manuscript PONE-D-24-57786, How do social network models compare to all-to-all models for forecasting tuberculosis epidemics? A mathematical modeling study”

Dear Dr. Mickael Essouma,

Thank you for the thoughtful and constructive feedback provided during the previous review rounds. We appreciate the opportunity to submit this second resubmission, in which we have addressed the most recent minor revision comments.

Below, we provide a point-by-point response. Reviewer comments are reproduced in italics, followed by our responses in plain text, with line numbers indicating where corresponding changes appear in the revised manuscript.

We are grateful for your continued guidance and careful consideration of our work.

Sincerely,

Masabho P. Milali, PhD

Comments

Comment 1: Regarding the EPIFORGE guidelines and to avoid redundancy, details in lines 186–191 and 200–205 need to be moved to the end of the introduction where the authors state the aim of the study.

Response: Thank you for this helpful suggestion. We have moved the information previously located in lines 186–191 and 200–205 to the end of the Introduction, as recommended. These changes are now reflected in lines 84–90.

Comment 2: There is a need to revise the presentation of figures in this manuscript. For instance, the footnote of Fig 1 describes panels A and B, but I do not see those panels on Fig 1. Line 36: is it Fig 1 or Fig 2? What are Figures 4 and 6?

Response: We appreciate this comment and have revised the figures for clarity and consistency.

• Figure 1 now clearly displays Panel A and Panel B, matching the caption.

• Figures 4 and 6 are now explicitly described in the text at lines 201–203 and 218–225 for Figure 4, and 292–306 for Figure 6.

• Regarding line 36, this line appears in the abstract, which does not contain figure citations. Figure 1 is described in lines 99–110, and Figure 2 in lines 148–152.

Comment 3: Fig. 3, 4, 5, and 6 should be clearly specified each (including any intrinsic panels) with its footnote and title together as you did for Fig 1 or Fig 2.

Response: As suggested, we have revised and standardized the titles and footnotes for Figures 3, 4, 5, and 6, ensuring that each figure clearly specifies any intrinsic panels and follows the same structure as Figures 1 and 2. These updates appear in lines 165–171, 218–225, 283–290, and 297–306.

Comment 4: Could you specify the published literature you are mentioning in line 293?

Response: Thank you for this observation. The relevant published literature has now been explicitly cited in line 86, following your earlier suggestion to relocate the content from lines 186–191 and 200–205 to the end of the Introduction where we state the study aims.

Comment 5: Did the End TB by 2035 strategy inform the target figures for treatment and vaccination used in this study or were they set arbitrarily? This should be addressed in the discussion as well.

Response: Thank you for highlighting this important point. The intervention levels used in our analysis (vaccination coverage of 30–70% and treatment improvements of 20–50%) were not derived from the WHO End TB Strategy 2035 operational targets, which focus on long-term reductions in incidence, mortality, and catastrophic costs. Instead, these values were selected as exploratory ranges commonly used in transmission modeling studies to examine the influence of incremental improvements in prevention and treatment. Their purpose was to isolate how contact structure modifies intervention impact while maintaining comparability across scenarios. We have added a clear explanation of this rationale in the Discussion (lines 361–370).

---

## [Editor Report · Decision Letter 2]

6 Feb 2026

How do social network models compare to all-to-all models for forecasting tuberculosis epidemics? A mathematical modeling study

PONE-D-24-57786R2

Dear Dr. Milali,

We’re pleased to inform you that your manuscript has been judged scientifically suitable for publication and will be formally accepted for publication once it meets all outstanding technical requirements.

Kind regards,

Mickael Essouma, M. D.

Academic Editor

PLOS One
---

## [Editor Report · Acceptance letter]

PONE-D-24-57786R2

PLOS One

Dear Dr. Milali,

I'm pleased to inform you that your manuscript has been deemed suitable for publication in PLOS One. Congratulations! Your manuscript is now being handed over to our production team.

Kind regards,

on behalf of

Dr. Mickael Essouma

Academic Editor

PLOS One